# 1,2-Dihydroxyxanthone: Effect on A375-C5 Melanoma Cell Growth Associated with Interference with THP-1 Human Macrophage Activity

**DOI:** 10.3390/ph12020085

**Published:** 2019-06-04

**Authors:** Viviana Silva, Fátima Cerqueira, Nair Nazareth, Rui Medeiros, Amélia Sarmento, Emília Sousa, Madalena Pinto

**Affiliations:** 1ICBAS, Institute of Biomedical Sciences Abel Salazar, University of Porto, Rua Jorge Viterbo Ferreira, 228, 4050-313 Porto, Portugal; vivianasilva19@gmail.com; 2FP-ENAS Research Unit, UFP Energy, Environment and Health Research Unit, CEBIMED, Biomedical Research Center, Fernando Pessoa University, Praça 9 de Abril, 349, 4249-004 Porto, Portugal; fatimaf@ufp.edu.pt (F.C.); ruimedei@ipoporto.min-saude.pt (R.M.); assuncao@ufp.edu.pt (A.S.); 3LAQV/REQUIMTE, Laboratory of Microbiology, Department of Biological Sciences, Faculty of Pharmacy, University of Porto, Rua Jorge Viterbo Ferreira, 228, 4050-313 Porto, Portugal; naircampos@gmail.com; 4Molecular Oncology and Viral Pathology GRP - IC, Portuguese Institute of Oncology of Porto (IPO Porto), Rua Dr. António Bernardino de Almeida, 4200-072 Porto, Portugal; 5LPCC, Research Department - Portuguese League Against Cancer (LPPC - NRN), Estrada Interior da Circunvalação, 6657, 4200-172 Porto, Portugal; 6i3S, Institute for Research and Innovation in Health, University of Porto, Rua Alfredo Allen, 208, 4200-135 Porto, Portugal; 7IBMC, Institute for Molecular and Cell Biology, University of Porto, Rua Alfredo Allen, 208, 4200-135 Porto, Portugal; 8CIIMAR/CIMAR, Interdisciplinary Centre of Marine and Environmental Research, Terminal de Cruzeiros do Porto de Leixões, Av. General Norton de Matos s/n, 4450-208 Matosinhos, Portugal; 9Laboratory of Organic and Pharmaceutical Chemistry, Department of Chemical Sciences, Faculty of Pharmacy, University of Porto, Rua Jorge Viterbo Ferreira, 228, 4050-313 Porto, Portugal

**Keywords:** xanthones, melanoma, macrophages, cytokines

## Abstract

Xanthones have been suggested as prospective candidates for cancer treatment. 1,2- dihydroxyxanthone (1,2-DHX) is known to interfere with the growth of several cancer cell lines. We investigated the effects of 1,2-DHX on the growth of the A375-C5 melanoma cell line and THP-1 human macrophage activity. 1,2-DHX showed a moderate growth inhibition of A375-C5 melanoma cells (concentration that causes a 50% inhibition of cell growth (GI_50_) = 55.0 ± 2.3 µM), but strongly interfered with THP-1 human macrophage activity. Supernatants from lipopolysaccharide (LPS)-stimulated THP-1 macrophage cultures exposed to 1,2-DHX significantly increased growth inhibition of A375-C5 cells, when compared to supernatants from untreated LPS-stimulated macrophages or to direct treatment with 1,2-DHX only. 1,2-DHX decreased THP-1 secretion of interleukin-1β (IL-1β) and interleukin-10 (IL-10), but stimulated tumor necrosis factor-α (TNF-α) and transforming growth factor-β1 (TGF-β1) production. This xanthone also inhibited nitric oxide (NO) production by RAW 264.7 murine macrophages, possibly through inhibition of inducible NO synthase production. In conclusion, these findings suggest a potential impact of 1,2-DHX in melanoma treatment, not only due to a direct effect on cancer cells but also by modulation of macrophage activity.

## 1. Introduction

Melanomas originate from melanocytic cells and are a result of multistep tumorigenesis [1]. Growing evidence suggests that the involvement of the immune system in the tumorigenic process is an important factor for malignancy progression [1,2]. Thus, investigation has focused on therapies directed at immune targets, resulting in the main approved therapeutic agent for melanoma used nowadays, ipilimumab (human monoclonal antibody (IgG1_k_) against the cytotoxic T-lymphocyte-associated antigen-4 (CTLA-4)) [3,4,5]. Nevertheless, the efficacies of both standard and alternative actual therapies are, in some cases, time-limited due to resistance mechanisms and, in other cases, associated with severe side effects [4,6,7]. Indeed, there is still a need to find more efficient and safe alternative treatments, preferably targeting multiple pathways, in order to prevent refractory tumors and consequently avoid the massive mortality rate associated with melanomas [5,8,9].

Investigation concerning xanthonic compounds has been rising, mostly due to their remarkable characteristics as potential drugs [10,11,12,13]. Xanthones from natural origins are, in fact, very promising compounds, but are limited in the positions or types of substituents in the xanthone scaffold due to the natural biosynthetic pathways [14,15]. The synthesis of new xanthones is important to enlarge the chemical diversity of these compounds and increase the possibility of finding new biological activities [11,16,17,18,19]. In a study with the human melanoma cell line UACC-62, although the unsubstituted xanthones had no cytotoxic effects on the cancerous cells, the presence of oxygenated substituents produced a growth inhibitory effect that was particularly promising for 1,2-dihydroxyxanthone (1,2-DHX) (Figure 1), which also showed some selectivity for melanoma cells when compared to MCF-7 (human breast cancer cell line) and TK-10 (human renal cancer cell line) cells. 1,2-DHX also showed a growth inhibitory effect on T-lymphocyte proliferation [15,20]. Moreover, this derivative was identified as a hit antioxidant compound, characterized by its chelating properties and its effects on a human keratinocyte cell line [21]. In another study by our group of researchers, chiral derivatives of xanthones also exhibited dose-dependent inhibitory effects on A375-C5 human melanoma cell lines [14]. Thus, xanthones are being described as interfering with the immune system, namely at the levels of proliferation of peripheral blood mononuclear cells (PBMC), cytokine production, or macrophage activity [15,22,23,24,25]. However, to the best of our knowledge, 1,2-DHX interference with macrophage activity has never been evaluated.

Macrophages are dynamic and heterogeneous cells, mainly due to their capacity to respond to stimuli. According to the microenvironment, they may be polarized into a spectrum of phenotypes, ranging from the pro-inflammatory M1 (classic) to the immunossupressive M2 (alternative) [26,27]. Several lines of evidence indicate that macrophage phenotypes can change during tumor progression [28,29,30]. M1 activation may induce chronic inflammation which can promote mutagenesis, cell proliferation and, consequently, predispose cells to tumor initiation [31,32]. In the early stages of tumor progression, tumor-associated macrophages (TAMs) adopt an M1-like phenotype that contributes to tumor immunity [33]. The M2 phenotype is mainly expressed in established tumors and induces immunosuppressive, angiogenic, and metastatic effects [30,34,35,36,37]. TAMs are important in melanoma therapies, not only because they are the most abundant immune cells present in the tumor but also because melanocytic tumors are described as highly immunogenic [28,38,39,40,41,42,43,44]. 

This work evaluated the activity of 1,2-DHX on the growth of the A375-C5 melanoma epithelial cell line and also its modulatory effects on the activity of the THP-1 macrophage cell line, namely cytokine production. The effects on the production of nitric oxide by the murine macrophage cell line RAW 264.7 were also investigated.

## 2. Results

### 2.1. 1,2-DHX and Supernatants from 1,2-DHX-Treated Macrophage Cultures Inhibited A375-C5 Growth

Incubation with 1,2-dihydroxyxanthone (1,2-DHX) resulted in a moderate inhibitory effect on A375-C5 melanoma cell growth (Table 1). 

The potential influence of 1,2-DHX-conditioned macrophage supernatants on A375-C5 cell growth was also investigated. The addition of supernatants obtained from THP-1 macrophages stimulated with lipopolysaccharide (LPS) to melanoma cell cultures resulted in growth inhibition comparable to that obtained by adding 1,2-DHX at a concentration of 50 μM (31.6 ± 4.9% and 25.3 ± 1.8%, respectively). However, the addition of supernatants from LPS-stimulated macrophages treated with 1,2-DHX to melanoma cells resulted in a significantly higher (p < 0.05) inhibitory effect on melanoma cell growth (52.4 ± 4.3%) than the addition of culture supernatants from macrophages stimulated with LPS only or the direct addition of xanthone at 50 μM, as demonstrated in Figure 2. The viability of THP-1 cells stimulated with LPS was determined, and was always ≥90% when compared to untreated LPS-stimulated control cells.

### 2.2. 1,2-DHX Modulated Cytokine Production by Macrophages

Further experiments were carried out to determine whether 1,2-DHX affected cytokine production by macrophages. Two concentrations of the compound (50 and 100 µM) were studied to evaluate the effects of this xanthone on cytokine production by an LPS-stimulated THP-1 macrophage cell line. The expressions of four cytokines were evaluated: Interleukin-1β (IL-1β) and tumor necrosis factor-α (TNF-α) (associated with M1 phenotype), and interleukin-10 (IL-10) and transforming growth factor-β1 (TGF-β1) (associated with M2 phenotype).

1,2-DHX significantly inhibited the expression of IL-1β and stimulated the expression of TNF-α at both concentrations (Figure 3). At the maximum concentration tested, 1,2-DHX significantly stimulated the expression of TGF-β1 and suppressed IL-10 production.

### 2.3. 1,2-DHX Inhibited NO Production by Macrophages

Inducible nitric oxide synthase (iNOS) mRNA expression was detected in RAW 264.7 murine macrophages stimulated with LPS, 2–4 h after stimulation, reaching the maximum at the time point of 6 h [45,46,47]. iNOS protein was detectable after 4–6 h of LPS stimulation in RAW 264.7 macrophages [45,46,47]. To determine whether the inhibitory effects were due to decreased enzyme synthesis or decreased enzyme activity, 1,2-DHX was added to RAW 264.7 at three different time points: Simultaneously with LPS, or 6 or 14 h after macrophage stimulation by LPS. In all cases, NO production was determined after a total incubation period of 24 h. 

To evaluate the effects of 1,2-DHX on NO production by the RAW 264.7 murine macrophage cell line, serial 1:2 dilutions were tested, and one concentration was selected in order to obtain an inhibition near 50% of NO production. The same procedure was used for *N*-nitro-L-arginine methyl ester (L-NAME) and dexamethasone, as positive controls. The concentrations selected were 25 μM for 1,2-DHX, 62.5 μM for L-NAME, and 6.25 μM for dexamethasone. 

When 1,2-DHX was added simultaneously with the stimulus, 56.6 ± 1.8% of NO production was inhibited. When the compound was added 6 h after stimulation, the inhibitory effect significantly decreased to 25.0 ± 2.8% (p < 0.001), and a lack of inhibitory effect was observed when the compound was added 14 h after stimulation (Table 2).

Toxicity was excluded by performing the thiazolyl blue tetrazolium bromide (MTT) viability assay, since cells showed viabilities higher than 90% at the dilution closest to IC_50_ of 1,2-DHX. 1,2-DHX did not show any scavenging activity of NO generated in a cell-free system.

## 3. Discussion

The anticancer and immunomodulatory effects of natural and synthetic xanthones have been extensively reported in the literature [10,13,24,48,49,50,51,52], including those of 1,2-DHX [15,20]. Despite these insights, as far as we know, no studies have yet been undertaken to clarify the effects of 1,2-DHX on macrophages, which may have an impact on melanoma cell growth.

In the present study, the effects of 1,2-DHX addition to cultures of the A375-C5 melanoma cell line were evaluated. We found that 1,2-DHX is a weaker inhibitor of A375-C5 cell growth (GI_50_ = 55.4 ± 2.0 µM) than of UACC-62 cell growth (GI_50_ = 14.0 ± 0.3 µM), another melanoma cell line [15]. This discrepancy may possibly be explained by the difference in cell type, since A375-C5 is of epithelial origin while UACC-62 is not.

Melanomas are classified as highly immunogenic tumors [1]. It was thus important to evaluate whether the anticancer activity of 1,2-DHX occurs through immune system involvement. We demonstrated that supernatants from THP-1 macrophages incubated with both LPS and 1,2-DHX strongly inhibited the growth of A375-C5 melanoma cells, when compared to the supernatants of THP-1 macrophages treated with LPS only. The mechanism underlying this effect is still unknown and deserves future evaluation.

Since cytokines produced by macrophages may play important roles in tumor growth or regression in vivo, we next evaluated the effects of 1,2-DHX treatment on cytokine production by THP-1 macrophages stimulated with LPS, which could account for the inhibition of A375-C5 growth induced by culture supernatants. We evaluated the productions of two cytokines associated with the M1 phenotype (IL-1β and TNF- α) and two others associated to the M2 phenotype (TGF-β1 and IL-10) [33]. Although cytokine production is time- and cell-dependent [53], cytokine secretion was quantified 24 h after LPS stimulation, since NO production was also assessed at that time point.

TNF-α levels were similar in LPS-stimulated and non-stimulated THP-1 cells, which is in accordance with the literature [53]. 1,2-DHX increased the levels of TNF-α and TGF-β1 detection in supernatants from LPS-stimulated THP-1 macrophages, while the detected levels of IL-1β and IL-10 were lower after xanthone exposure. The stimulatory effect of 1,2-DHX on TNF-α production predicts a favorable outcome in melanoma treatment, since several reports have associated the use of TNF inhibitors with an increased risk of developing skin cancer, including melanoma [54,55]. However, this idea is not corroborated by recent studies [56,57]. The A375 melanoma cell line expresses low levels of TNF-α, and this cytokine, derived from myeloid cells, was described as fundamental for melanoma growth in vitro [57].

IL-1β, a pro-inflammatory cytokine mainly produced by monocytes and macrophages, is an example of immune system pleiotropism. In melanoma, its expression was associated to tumor progression and promotion of lung metastases from melanomas [58,59]. Thus, IL-1β has been associated with all steps of malignancy (carcinogenesis, progression, invasion, and metastasis) and may even be expressed by the tumor cells [60]. In contrast, it was also found to induce an immune response against malignant cells associated to the M1 macrophage phenotype [61]. However, the A375-C5 cell line is resistant to the effects of IL-1α [62,63,64]. 

IL-10 and TGF-β1 are anti-inflammatory cytokines involved in the carcinogenesis process. In melanomas, IL-10 was associated with metastatic formation [65,66], and TGF-β is highly expressed and increases in parallel with tumor progression [67,68]. The treatment of LPS-stimulated macrophages with 1,2-DHX might be expected to affect the expressions of the cytokines selected to represent the M1 (IL-1β and TNF-α) and M2 (TGF-β1 and IL-10) macrophage profiles in a similar way. Nevertheless, the results showed that the detected levels of some M1 and M2 characteristic cytokine expressions increased (TNF-α and TGF-β1, respectively), and others decreased (IL-1β and IL-10, respectively). 

Taking these findings together, the 1,2-DHX-dependent increase in TNF-α and decrease in IL-10 detection may contribute to tumor regression, although the TGF-β1 increase seems to contradict the xanthone antitumor effects. However, neither IL-10 nor TGF-β1 were probably involved in the antitumor effects of the 1,2-DHX-conditioned THP-1 culture supernatant, since at the xanthone dose used (50 µM), no significant changes were observed for IL-10 or TGF-β1 levels. In conclusion, the isolated analysis of the alterations on cytokine levels seems to be insufficient to explain the antitumor effects observed for 1,2-DHX-conditioned LPS-stimulated THP-1 supernatants, suggesting a multifactorial interference responsible for the aforementioned results. This is in agreement with other studies, in which the effects of macrophage-derived factors in tumor growth, namely melanomas, are the result of interactions of factors with tumor-promoter and tumor-inhibitory activities [57]. 

The role of nitric oxide (NO) produced by macrophages in tumors is controversial. It was reported that below a critical concentration, NO causes DNA mutations [69], inhibits apoptosis [70], promotes angiogenesis [71], limits immune responses against cancer [72], and promotes metastasis [73]. However, when it exceeds the critical concentration, NO induces apoptosis and suppresses the growth of the tumor [74], and is reported as being cytotoxic in melanomas and many other tumors [74,75]. With the purpose of evaluating the effects of 1,2-DHX on NO production, xanthone was added at different time points: 0 (simultaneously with LPS), 6, and 14 h after RAW 264.7 macrophage stimulation with LPS. 1,2-DHX is a potent inhibitor of NO production by LPS-stimulated RAW 264.7 murine macrophage cells, and this effect was not due to cell death or to NO scavenging. However, when 1,2-DHX is added 6 h after stimulation, the inhibitory effect decreased significantly (p < 0.001), and when added after 14 h, no inhibition was detected. These findings suggest that the inhibitory effect of 1,2-DHX on NO production may depend on inhibition of iNOS expression, since at 6 h after LPS stimulation, iNOS expression has already occurred. L-NAME (an inhibitor of iNOS activity) and dexamethasone (an inhibitor of iNOS expression) were used as positive controls of inhibition [76,77]. 1,2-DHX showed comparable results to dexamethasone, which is in favor of an inhibitory effect of xanthone on iNOS expression.

The inhibition of iNOS expression indicates that 1,2-DHX has anti-inflammatory properties. Inhibition of iNOS expression may be related to the increase of TGF-β1 concentration. This cytokine has been reported as a destabilizer of iNOS mRNA, avoiding gene translation and inhibiting NO production. The NF-κB pathway may also be involved, since it contributes to iNOS synthesis and upregulation of IL-1β levels. 1,2-DHX-treated LPS-stimulated macrophages expressed lower IL-1β and NO levels compared to untreated LPS-stimulated macrophages (THP-1 and RAW 264.7, respectively). These findings lead us to hypothesize the interference of the NF-κB pathway by xanthone. More studies must be performed to corroborate this possibility and elucidate the mechanism of macrophage modulation responsible for the antitumor effect.

In conclusion, 1,2-DHX may be a promising compound for melanoma treatment, due to its direct inhibition of melanoma cell growth and modulation of macrophage activity. The lack of appropriate current therapeutic alternatives to fight against melanomas makes the studies on new compounds highly needed and emphasizes the importance of these results. Since the immune response is crucial for tumor regression or growth, further studies are required to clarify the mechanisms underlying the effects of 1,2-DHX on macrophages and to evaluate effects on other immune system cells.

## 4. Materials and Methods

### 4.1. Xanthone

1,2-DHX was previously synthetized by our group of researchers [16]. A stock solution was prepared in DMSO (Applichem, Darmstadt, Germany) and, before each assay, diluted in the appropriate complete culture medium.

### 4.2. Cell Lines

The THP-1 human monocyte cell line, the A375-C5 human malignant melanoma cell line, and the RAW 264.7 mouse macrophage cell line were a courtesy of Rui Appelberg (Immunobiology Group, IBMC, Porto, Portugal), Helena Vasconcelos, and Maria de São José Nascimento (Laboratory of Microbiology, Department of Biological Sciences, Faculty of Pharmacy, Porto, Portugal), respectively. A375-C5 was routinely maintained in 25 cm^2^ culture flasks containing Roswell Park Memorial Institute-1640 (RPMI-1640) medium with glutamax (Gibco, Paisley, UK), supplemented with 10% fetal bovine serum (FBS; Gibco, Paisley, UK) and 50 µg mL^−1^ gentamycin (Sigma-Aldrich, St. Louis, MO, USA) at 37 °C in a humidified incubator with 5% CO_2_. For the THP-1 cell line, the culture medium just described was also supplemented with 0.05 mM 2-mercaptoethanol (Merck, Darmstadt, Germany). The RAW 264.7 cell line was maintained in 75 cm^2^ flasks containing Dulbecco’s Modified Eagle Medium (DMEM) with glutamax (Gibco, Paisley, UK), with 10% FBS and 50 µg mL^−1^ gentamycin, in the same conditions.

### 4.3. Sulphorodamine B (SRB) Growth Inhibition Assay

The effects of 1,2-DHX on the growth of the A375-C5 human melanoma cell line (7.5 × 10^4^ cells mL^−1^) were evaluated according to the method adopted by the National Cancer Institute (NCI, Oxford, MS, USA) [78] as already described by our group [16,79]. Doxorubicin (1:10 dilutions; Sigma-Aldrich, St. Louis, MO, USA) was used as a positive control [16,79]. Results were expressed as the concentration that causes a 50% inhibition of cell growth (GI_50_) when compared to the control.

### 4.4. Antitumor Effect of Conditioned Macrophage Culture Medium

The antitumor effect of 1,2-DHX-conditioned macrophage culture medium on the A375-C5 cell line was evaluated according to [80]. Briefly, the THP-1 cell line was plated at 1 × 10^5^ cells mL^−1^ and differentiated into THP-1 macrophages with 10 ng mL^−1^ phorbol 12-myristate 13-acetate (PMA) (Sigma-Aldrich, St. Louis, MO, USA) for 72 h [80]. Once differentiated, cells were washed twice with complete medium and left for another 24 h incubation in order to achieve a resting state [81,82]. Then, they were stimulated with 100 µL of LPS solution (1 µg mL^−1^; Sigma-Aldrich, St. Louis, MO USA) and treated, or not, with 100 µL of 1,2-DHX solution. Cells were further incubated for 24 h at 37° C, 5% CO_2_ in a humidified incubator [80]. Plates were centrifuged, and half of the volume of each well was transferred to A375-C5 adherent cell monolayers, previously plated as for the cancer growth inhibition assay procedure. After 48 h incubation, an SRB assay was performed, and absorbance was measured and the percentage of cell-growth inhibition determined as compared to the control cells [16].

### 4.5. Cytokine Quantification

Culture supernatants of differentiated THP-1 cells, unstimulated and LPS-stimulated, treated or not, and with 1,2-DHX as previously described, were stored at −20 °C until cytokine analyses. Levels of IL-1β, IL-10, TNF-α, and TGFβ1 in culture supernatants were quantified by ELISA Ready-Set-Go Kits (eBioscience, San Diego, CA, USA) according to the manufacturers’ instructions.

### 4.6. NO Production Assays

NO production by RAW 264.7 cells (1 × 10^6^ cells mL^−1^) after LPS stimulation (1.5 µg mL^−1^) was quantified by Griess assay, as previously described by our group [24,82]. The inhibition of NO production was determined in terms of percentage, in relation to the NO produced by the LPS-stimulated control cells (100% production). L-NAME and dexamethasone (both from Sigma-Aldrich, St. Louis, MO, USA) were used as positive controls. 

### 4.7. NO Scavenging Assays

To assess NO scavenging effects by xanthone, nitrite was chemically generated using sodium nitroprusside (Sigma-Aldrich, St. Louis, MO, USA), and quantified by Griess assay as previously described [25,83].

### 4.8. MTT-Viability Assay

The effects of 1,2-DHX on THP-1 and RAW 264.7 cell viability were evaluated by the MTT (thiazolyl blue tetrazolium bromide) viability assay [16,25,84].

### 4.9. Statistical Analysis

Except otherwise stated, results are the mean ± SEM of at least three independent experiments, performed in duplicate. Statistical analysis was performed with SPSS for Windows (version 20.0). Statistical significance between groups was calculated by the Mann–Whitney Test, and it is considered for p values less than 0.05.

## Figures and Tables

**Figure 1 pharmaceuticals-12-00085-f001:**
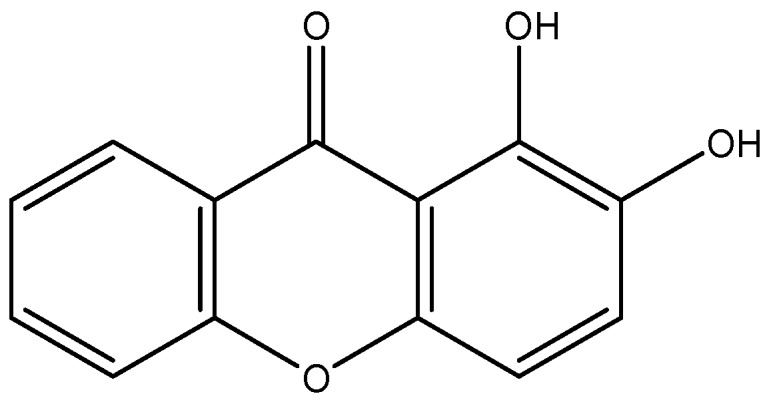
Structure of 1,2-dihydroxyxanthone (1,2-DHX).

**Figure 2 pharmaceuticals-12-00085-f002:**
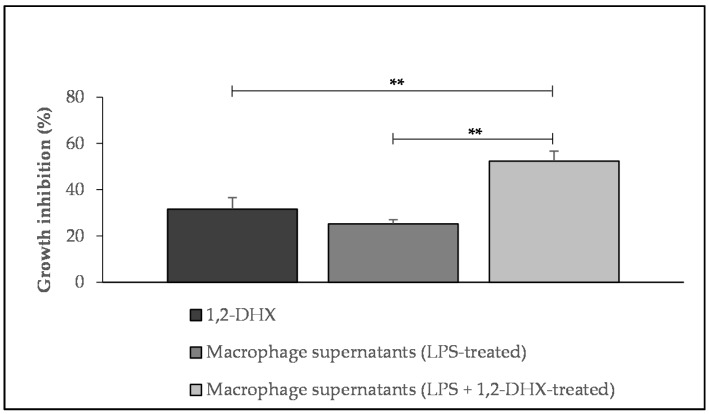
Effect of 1,2-dihydroxyxanthone (1,2-DHX; 50μM), lipopolysaccharide (LPS)-stimulated macrophage supernatants, and 1,2-DHX-conditioned LPS-stimulated macrophage supernatants on A375-C5 melanoma cell line growth. Results show mean values ± SEM (n ≥ 3). ** p < 0.05.

**Figure 3 pharmaceuticals-12-00085-f003:**
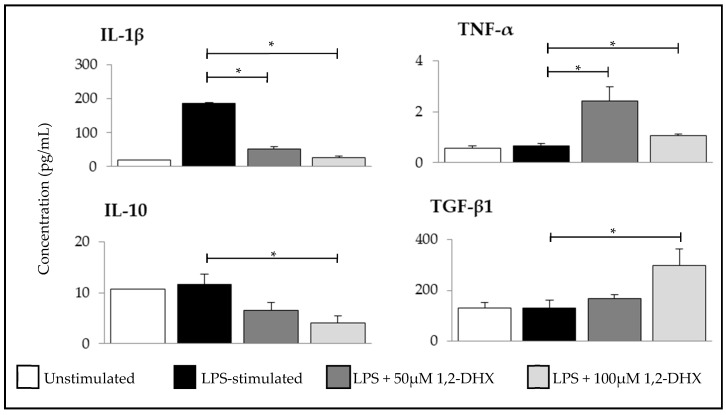
Effects of 1,2-dihydroxyxanthone (1,2-DHX) on IL-1β, IL-10, TGF-β1, and TNF-α production by THP-1 macrophages. Cytokine production was evaluated after 24 h incubation with unstimulated macrophages (basal), LPS-stimulated macrophages (positive control), and LPS-stimulated macrophages treated with 1,2-DHX (50 or 100 µM). Data are the mean ± SEM from one experiment, performed with duplicate cultures, and are representative of three experiments carried out independently. * p < 0.001.

**Table 1 pharmaceuticals-12-00085-t001:** Effect of 1,2-dihydroxyxanthone on the growth of the A375-C5 human melanoma cell line.

Compound	Growth inhibition (GI_50_)
1,2-DHX	55.4 ± 2.0 µM
Doxorubicin	1.8 × 10^−2^ ± 0.4 × 10^−2^ µM

Results are the mean ± SEM (standard error of the mean) (n ≥ 3). Doxorubicin was used as a positive control. 1,2-DHX: 1,2-Dihydroxyxanthone. GI_50_: Concentration that was able to cause 50% cell growth inhibition.

**Table 2 pharmaceuticals-12-00085-t002:** Inhibitory effect of 1,2-dihydroxyxanthone (1,2-DHX) on nitric oxide (NO) production by RAW 264.7 cells.

Compound	NO Inhibition (% of Control)		
	0 h	6 h	14 h
1,2-DHX (25 μM)	56.6 ± 1.8	25.0 ± 2.8 *	n.i.
L-NAME (62.5 μM)	52.8 ± 5.0	50.7 ± 3.8	25.9 ± 2.7 **
Dexamethasone (6.25 μM)	55.9 ± 2.3	15.7 ± 4.9 *	n.i.

Macrophages were exposed to LPS and treated with 1,2-dihydroxyxanthone (1,2-DHX) at different times after stimulation: 0 (simultaneously with the stimulus), 6, or 14h after stimulation. Results are the mean ± SEM (n ≥ 3). n.i. = no inhibition. * p < 0.001, ** p < 0.05. *N*-nitro-L-arginine methyl ester (L-NAME) and dexamethasone were used as positive controls.

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
