# Peer review of "1,2-Dihydroxyxanthone: Effect on A375-C5 Melanoma Cell Growth Associated with Interference with THP-1 Human Macrophage Activity"

_pharmaceuticals, 2019, doi:10.3390/ph12020085_

Round 1
Reviewer 1 Report
There is yet a sentence in the abstract that does not correspond to the results shown in Fig. 2.
Line 37 (abstract). The authors state:
Supernatants from THP-1 macrophages cultures exposed to 1,2-DHX significantly increased growth inhibition of A375-C5 cells, when compared to supernatants from untreated macrophages or to direct treatment with 1,2-DHX alone.
But according to the results shown in Fig 2, the term “untreated macrophages” does not apply. The sentence should include that THP1 macrophages were always treated with LPS. For instance:
Supernatants from THP-1 macrophages cultures exposed to 1,2-DHX plus LPS significantly increased growth inhibition of A375-C5 cells, when compared to supernatants from LPS-treated macrophages or to direct treatment with 1,2-DHX alone.
Author Response
Comment - “There is yet a sentence in the abstract that does not correspond to the results shown in Fig. 2. Line 37 (abstract). The authors state: Supernatants from THP-1 macrophages cultures exposed to 1,2-DHX significantly increased growth inhibition of A375-C5 cells, when compared to supernatants from untreated macrophages or to direct treatment with 1,2-DHX alone. But according to the results shown in Fig 2, the term “untreated macrophages” does not apply. The sentence should include that THP1 macrophages were always treated with LPS. For instance: Supernatants from THP-1 macrophages cultures exposed to 1,2-DHX plus LPS significantly increased growth inhibition of A375-C5 cells, when compared to supernatants from LPS-treated macrophages or to direct treatment with 1,2-DHX alone.”
We thank the reviewer for identifying the lapse and the sentence was corrected.
Reviewer 2 Report
The authors have provided a revised version of the paper “Effect of 1,2-dihydroxyxanthone on A375-C5 melanoma cell growth and on THP-1 macrophages mediated tumor microenvironment" which now has the title “1,2-Dihydroxyxanthone: effect on A375-C5 melanoma cell growth associated with interference with THP-1 human macrophages activity”.
The authors have partially replied to referees' objections and put off the evaluation of iNOS expression in a future study.
However, the manuscript is now sufficiently improved.
I wish to invite the authors to give attention to the following points:
- Abstract, lines 34-35, please correct: We investigated the 1,2-DHX effect on the growth of A375-C5 melanoma cell line and THP-1 human macrophages activity.”
- The authors should cite the Figure 1 in the text, probably in the Introduction.
- Lines 112-113-, delete **
Author Response
Comment –“Abstract, lines 34-35, please correct: We investigated the 1,2-DHX effect on the growth of A375-C5 melanoma cell line and THP-1 human macrophages activity.”
We agree with the reviewer and the change was made as suggested.
Comment – “The authors should cite the Figure 1 in the text, probably in the Introduction.”
The authors proceeded as suggested.
Comment – “Lines 112-113-, delete **”
The authors proceeded as suggested.
Reviewer 3 Report
The study by Silva V. et al. focuses on the influence of 1,2-dihydroxyxanthone on the growth of A375-C5 melanoma cells, the activity of THP-1 human macrophages and nitric oxide production by RAW 264.7 murine macrophages. Only some of comments have been properly addressed in the revised version of the manuscript.
Major comments:
Comment 1
Authors performed experiments using only one melanoma cell line (A375-C5), which is not sufficient to draw conclusions presented in this report. Moreover, using A375-C5 cells is not well justified in this report. Authors did not address these comments properly.
Comment 2
Properly addressed, however, in the sentence: “1,2-DHX treated macrophages expressed either lower IL-1β and NO levels comparatively to untreated LPS-stimulated macrophages” the word “either” should be removed.
Comment 3
Authors should more broadly discuss the phenotypes of macrophages considering changes in the cytokine concentration after LPS stimulation and treatment with 1,2-DHX.
Comment 4
Authors correctly defined Y-axis in the Figure 1 as well as in the title of Table 2.
Comment 5
Authors slightly improved the Introduction that was enriched with the article of Wang et al. 2017 [2].
Minor comments:
Comment 1: authors properly defined all abbreviations.
Comment 2: authors added the chemical formula of 1,2-DHX (Figure 1) in the Introduction.
Comment 3: authors corrected unnecessarily underlined abbreviation of Celsius degrees.
However, the manuscript still contains several mistakes e.g.:
Line 103: Addition to melanoma cell cultures of supernatants
Line 165: (…), wich may have impact in melanoma (…)
Line 167: We found that 1,2-DHX is a weeker
Line 199: IL-10 and TGF-ß are anti-inflammatory cytokine (…)
Line 206: no significantly changes were observed
Line 229: This find leads us to (…)
Line 235: current therapeutic alternatives to fight melanoma (…) etc.
For that reason, the manuscript would certainly benefit from extensive editing by a native speaker of English.
Author Response
Major comments:
Comment 1 – “Authors performed experiments using only one melanoma cell line (A375-C5), which is not sufficient to draw conclusions presented in this report. Moreover, using A375-C5 cells is not well justified in this report. Authors did not address these comments properly.”
In a first study on the effect of 1,2-DHX on the growth of human cancer cell lines, UACC-62 melanoma cell line was used. As described in introduction, other xanthone derivatives effect on the growth of A375-C5 melanoma cell line was addressed by our group. In the present study, we decided to study the effect of 1,2-DHX on the growth of A375-C5 cancer cell line, to extend, since we wanted to extend the panel considering the effect on the growth of melanoma cell lines, since this was already the second report concerning that effect. In addition, has we wanted to evaluate the effect of 1,2-DHX on macrophages activity, the same cell line was used (A375-C5 cell line). The characteristics of the cell line were also reported.
Following this explanation, and beyond, the authors find no way to clarify the referees' doubts.
Comment 2 – “Properly addressed, however, in the sentence: “1,2-DHX treated macrophages expressed either lower IL-1β and NO levels comparatively to untreated LPS-stimulated macrophages” the word “either” should be removed.”
The change was made as suggested. The word “either” could induce the effect to the same cell line. So, besides removing the word, and to better clarify the sentence, adding (Raw264.7 and THP-1, respectively). The final sentence is now:
“1,2-DHX treated LPS-stimulated macrophages expressed lower IL-1β and NO levels comparatively to untreated LPS-stimulated macrophages (Raw264.7 and THP-1, respectively).”
Comment 3 – “Authors should more broadly discuss the phenotypes of macrophages considering changes in the cytokine concentration after LPS stimulation and treatment with 1,2-DHX.”
A new paragraph was included in discussion.
Minor comments:
Comment: “Line 103: Addition to melanoma cell cultures of supernatants”
Correction was performed.
Comment: “Line 165: (…), wich may have impact in melanoma (…)”
Correction was performed and the sentence is now: “(…) which may have an impact on melanoma (…)”
Comment – “Line 167: We found that 1,2-DHX is a weeker”
Correction was performed. “We found that 1,2-DHX is a weaker”
Comment – “Line 199: IL-10 and TGF-ß are anti-inflammatory cytokine (…)”
Correction was performed. “IL-10 and TGF-ß are anti-inflammatory cytokines
Comment – “Line 206: no significantly changes were observed”
Correction was performed. “no significant changes were observed”
Comment – “Line 229: This find leads us to (…)”
Correction was performed. “These findings lead us”
Comment – “Line 235: current therapeutic alternatives to fight melanoma (…) etc.
Correction was performed. “to fight against melanoma”
As suggested, an extensive editing of the English language was performed.
Round 2
Reviewer 3 Report
Major comments:
Comment 1:
Authors still did not addressed a major comment 1. Moreover, their explanation is not convincing.
Comment 2:
Authors properly addressed this comment and removed the word “either”, however, another misleading information was introduced. I think, IL-1β expression was investigated using THP-1 macrophages, whereas NO production was evaluated in RAW 264.7 murine macrophages.
Comment 3:
Authors tried to address this comment, however, description that is provided in improved version of the manuscript (lines 206-210) should be re-written.
Minor comments:
Authors properly addressed all minor comments. However, still spelling and grammatical errors could be found in the manuscript (e.g., in line 81 “may induces”). As this mistake (“may induces”) was introduced as a part of “an extensive editing of the English language” declared by the Authors, it might be necessary to ask somebody more fluent in English to check the language of the manuscript.
Author Response
Major comments:
Comment 1: “Authors still did not addressed a major comment 1. Moreover, their explanation is not convincing.”
The authors tried to address the comment of the reviewer and previously presented their justification. Probably the authors did not realize exactly what the reviewer concern was.In the quest to fulfill the reviewer comment, the authors found a reference in which TNF-a derived from myeloid cells was identified as a growth promoter of melanoma cells (Smith et al., 2014). In the same study, the researchers refer that A375 cell line produces low levels of TNF-a. In our study, TNF-a expression by LPS-stimulated THP-1 macrophages was increased by 1,2DHX treatment. Despite that, supernatants from 1,2-DHX treated LPS-THP1 macrophages significantly inhibited the growth of tumor cancer cell lines. Therefore additional information and the folllowing reference was included in the revised manuscript (lines 190-192 and 214-219).
Smith, M., Sanchez-Laorden, B., O’Brien, K., Brunton, H., Ferguson, J., Young, H., Dhomen, N., Flaherty, K., Frederick, D., Cooper, Z., Wargo, J., Marais, R., and Wellbrock, C. The immune-microenvironment confers resistance to MAP kinase pathway inhibitors through macrophage-derived TNFα. Cancer Discov. 2014, 4, 1214–1229. doi:10.1158/2159-8290.CD-13-1007.
Comment 2: “Authors properly addressed this comment and removed the word “either”, however, another misleading information was introduced. I think, IL-1β expression was investigated using THP-1 macrophages, whereas NO production was evaluated in RAW 264.7 murine macrophages. “
We thank the reviewer for identifying the lapse and the sentence was corrected.
Comment 3: “Authors tried to address this comment, however, description that is provided in improved version of the manuscript (lines 206-210) should be re-written.”
The paragraph included in the last version of the manuscript was re-written, as suggested.
Minor comments:
“Authors properly addressed all minor comments. However, still spelling and grammatical errors could be found in the manuscript (e.g., in line 81 “may induces”). As this mistake (“may induces”) was introduced as a part of “an extensive editing of the English language” declared by the Authors, it might be necessary to ask somebody more fluent in English to check the language of the manuscript.”
A new editing of the English language was performed.
This manuscript is a resubmission of an earlier submission. The following is a list of the peer review reports and author responses from that submission.
Round 1
Reviewer 1 Report
In this report Silva et al. study the effect of a xanthone on the growth of a melanoma cell line. This could be of potential interest for cancer treatment. However, there are several limitations to this work:
· The study is on a particular melanoma cell line only, and the conclusions derived cannot be easily generalized.
· Talking of tumor microenvironment in the title and abstract is misleading. There are no in vivo experiments in this study related to tumor microenvironment, and there are no in vitro experiments trying to reproduce a tumor microenvironment. At most, the supernatants of THP1 macrophage leukemic cells, but not primary macrophages, are incubated with melanoma cells.
· The significance of the results obtained using leukemic THP1 cells instead of primary cells is unclear. Why the authors have not used primary macrophages?
· Table I is not informative enough. The complete inhibitory curve of 1,2-DHX should be drawn, presenting the maximum percentage of growth inhibition reached with 1,2-DHX. Furthermore, is the inhibition of growth the result of direct toxicity of 1,2-DHX on melanoma cells? Or, contrarily, the inhibition of growth results from inhibition of proliferation, but cells remain viable?
· Figure 1. What is the effect of 1,2-DHX on THP1 viability? I wonder whether putative apoptosis of THP1 by 1,2-DHX could affect the growth of melanoma cells.
· Figure 1. The figure legend does not indicate that THP1 cells were LPS-treated, but the main text does (line 96). Was then LPS included in the experiment?
· Figure 1. A control point of melanoma cells treated with supernatant of unstimulated THP1 is lacking. As shown in Fig 2, unstimulated THP1 cells secrete IL-10 and could have a direct effect on melanoma cells.
· Figure 2. A control point of THP! cells treated only with 1,2-DHX without LPS is lacking. In fact, in melanoma tumors, macrophages are not expected to be activated by LPS.
· Finally, the mechanism of the inhibition of the growth of melanoma cells by THP1 supernatant has not been characterized. Have the authors tried to reproduce the effect of macrophage supernatant by adding recombinant TNF and TGFB to melanoma cells?
Minor points.
Page 2, line 48. The authors refer to the growing evidence of the involvement of the immune system in the tumorigenic process but cite two rather old references (1,2) from 2004 and 2011. Moreover, ref 1 provides no information sustaining the author’s statement that “the involvement of the immune system in the tumorigenic process is the main factor for malignancy progression”.
Page 3, line 88. The authors state: 1,2-Dihydroxyxanthone (1,2-DHX) resulted in a moderate inhibitory effect on A375-C5 melanoma cell growth (Table 1). Probably they wanted to say: Incubation with 1,2-Dihydroxyxanthone (1,2-DHX) resulted in a moderate inhibitory effect on A375-C5 melanoma cell growth (Table 1).
Reviewer 2 Report
In the paper entitled "Effect of 1,2-dihydroxyxanthone on A375-C5 melanoma cell growth and on THP-1 macrophages mediated tumor microenvironment", by Silva et al., the authors report data regarding the effects of 1,2-DHX on the growth of A375-C5 melanoma cell line and also the effects on the activity of the THP-1 macrophages.
The authors show that addition to A375-C5 cells of supernatants from LPS-stimulated macrophages treated with 1,2-DHX resulted in a high inhibition of melanoma cell growth. Moreover, 1,2-DHX modulated the cytokine (IL-1β, TNF-α, IL-10 and TGF-β1) production by LPS-stimulated THP-1 macrophages and inhibited NO production by RAW 264.7 cell line.
Taken together, the reported findings are interesting and could add further information on 1,2-DHX
that could be a promising compound for melanoma treatment.
However, the following issues should be addressed:
- The authors should show the chemical structure of 1,2- Dihydroxyxanthone (1,2-DHX)
- The authors should also report the effect of 1,2-DHX on the growth of other human cancer cell lines
- The authors should show the data of effect of 1,2-DHX on nitrite production by RAW 264.7 macrophages stimulated with LPS
- The authors should evaluate the effect of 1,2-DHX on the iNOS expression in LPS-stimulated RAW 264.7 macrophages
- In Table 2, correct the title “Inhibitory effect of 1,2-dihydroxyxanthone (1,2-DHX) on NO production by RAW 264.7 cells
- Pag. 6, line 187, correct “and increases”
- The authors should check the manuscript carefully.
Reviewer 3 Report
The study by Silva V. et al. focuses on the influence of 1,2-dihydroxyxanthone (1,2-DHX) on A375-C5 melanoma cell line, the activity of THP-1 human macrophages and nitric oxide production by RAW 264.7 murine macrophages. Obtained results suggest that 1,2-DHX affects melanoma cells either directly or by modulation of macrophage activity in the tumor microenvironment. The study reports exclusively in-vitro data in support of presented conclusions. Although the influence of a drug-modified microenvironment on cancer cells is of course a quite interesting issue, this aim is not satisfactorily achieved. The authors should properly address the following specific comments.
1. Conclusions are driven based on experiments performed with only one melanoma cell line (A375-C5), which is not sufficient. At least two additional melanoma cell lines should be used to assess the influence of 1,2-DHX. Moreover, using A375-C5 is not well justified in this report (see below).
2. While 1,2-DHX significantly inhibited the expression of IL-1β in macrophages stimulated by LPS, the level of IL-1β in macrophages treated with 1,2-DHX is not shown and it is not compared with control level. Therefore, the sentence in the Discussion “1,2-DHX treated macrophages expressed lower IL-1β (..) levels comparatively to control cells.” and the hypothesis that follows that sentence are not supported by the results or references. Moreover, A375-C5, used in the study, is a clonal variant line that is resistant to IL-1 alpha (Usui 1991).
3. It is not evident which phenotype of macrophages (polarized into M1 or M2) was obtained after LPS-stimulated THP-1 cells were treated with 1,2-DHX.
4. Please define “Growth inhibition (%)” on Y-axis in Figure 1 and “NO inhibition (%)” in Table 2. Description of the Methods and Results sections is not comprehensive enough to find it out. Therefore, it is unclear whether interpretation provided in the report is correct.
5. The Introduction does not provide sufficient background to experiments. I would also recommend a more rigorous description and discussion of the data.
Minor:
1. Some of abbreviations e.g., LPS, IL-10 etc. are explained in Materials and Methods, however they should be defined the first time they appear in the text.
2. The chemical formula of 1,2-DHX might be included.
3. In the section “Materials and methods” the abbreviation of Celsius degrees “°“ is unnecessarily underlined.